# Phosphorylation of Arp2 is not essential for Arp2/3 complex activity in fission yeast

Alexander E Epstein[1] , Sofia Espinoza-Sanchez[2], Thomas D Pollard[1,2,3]

**LeClaire et al presented evidence that phosphorylation of three sites on the Arp2 subunit activates the Arp2/3 complex to nucleate actin filaments. We mutated the homologous residues of Arp2 (Y198, T233, and T234) in the fission yeast genome to amino acids that preclude or mimic phosphorylation. Arp2/3 complex is essential for the viability of fission yeast, yet strains unable to phosphorylate these sites grew normally. Y198F/T233A/T234A Arp2 was only nonfunctional if GFP-tagged, as observed by LeClaire et al in *Drosophila* cells. Replacing both T233 and T234 with aspartic acid was lethal, suggesting that phosphorylation might be inhibitory. Nevertheless, blocking phosphorylation at these sites had the same effect as mimicking it: slowing assembly of endocytic actin patches. Mass spectrometry revealed phosphorylation at a fourth conserved Arp2 residue, Y218, but both blocking and mimicking phosphorylation of Y218 only slowed actin patch assembly slightly. Therefore, phosphorylation of Y198, T233, T234, and Y218 is not required for the activity of fission yeast Arp2/3 complex.**

## Introduction

Assembly of branched actin filament networks drives cellular processes including cell motility and clathrin-mediated endocytosis (Weinberg & Drubin, 2012; Blanchoin et al, 2014). The seven-subunit Arp2/3 complex builds these networks by binding to the side of a "mother" actin filament and nucleating a "daughter" filament branch (Mullins et al, 1998). Activation of the Arp2/3 complex depends on the binding of nucleation-promoting factors (NPFs) (Machesky & Insall, 1998; Machesky et al, 1999; Rohatgi et al, 1999; Winter et al, 1999; Yarar et al, 1999) which induce a conformational change (Hetrick et al, 2013; Espinoza-Sanchez et al, 2018) and promote binding of the complex to the side of a mother filament (Ti et al, 2011). For example, the NPF Wiskott–Aldrich syndrome protein (WASp) is recruited to sites of endocytosis where it activates the Arp2/3 complex (Winter et al, 1999). The Arp2/3 complex then builds a "patch" of branched actin filaments that generates force to internalize endocytic vesicles (Carlsson & Bayly, 2014). In motile cells, the SCAR/WAVE complex activates the Arp2/3 complex along the leading edge of the cell, stimulating the formation of the lamellipodium that sweeps the cell forward (Insall & Machesky, 2009). Regulation of the Arp2/3 complex is essential to control the localization and assembly of branched actin networks.

LeClaire et al (2008) discovered that purified *Acanthamoeba* Arp2/3 complex lost its ability to nucleate actin filaments when treated with serine/threonine and tyrosine phosphatases. Furthermore, antibodies to phosphothreonine and phosphotyrosine interacted with the Arp2 and Arp3 subunits of the Arp2/3 complex from *Acanthamoeba*, *Bos taurus*, and humans. Mass spectrometry was used to identify phosphorylation of the highly conserved T237 and T238 residues of amoeba Arp2. The location of Y202 near these threonines suggested that it might also be phosphorylated. LeClaire et al (2008) investigated the role of phosphorylation at Y202, T237, and T238 in regulating the *Drosophila* Arp2/3 complex. Depletion of Arp2 compromised the formation of lamellipodia in *Drosophila* S2 cells. This defect was rescued by the expression of wild-type Arp2-GFP, T237A/T238A Arp2-GFP, or Y202A Arp2-GFP, but not by Y202A/T237A/T238A Arp2-GFP, indicating that phosphorylation of either the two threonines or the tyrosine is essential for Arp2/3 complex activity. A *Drosophila* kinase that phosphorylates these residues has been identified: In 2015, LeClaire et al (2008) reported that the Nck-interacting kinase (NIK) can phosphorylate several Arp2/3 complex subunits, including Arp2 at Y202, T237, or T238 (LeClaire et al, 2015). NIK restored the actin nucleation activity of the purified Arp2/3 complex after the complex was inactivated by treatment with serine/threonine and tyrosine phosphatases.

LeClaire et al (2008) originally suggested that phosphorylation at Y202, T237, and T238 activates the Arp2/3 complex by disrupting inhibitory interactions of these residues with R409 of the Arp3 subunit and R105 and/or R106 of the ARPC4 subunit. A 2011 study used molecular dynamics (MD) simulations to study the effects of the interactions involving these phosphorylated residues on the structure of the Arp2/3 complex (Narayanan et al, 2011). During all-atom MD simulations of the native Arp2/3 complex for 30 ns, Arp2 shifted 3–4 Å relative to Arp3 from its position in the inactive crystal

[1]Departments of Molecular Cellular and Developmental Biology, Yale University, New Haven, CT, USA   [2]Departments of Molecular Biophysics and Biochemistry, Yale University, New Haven, CT, USA   [3]Department of Cell Biology, Yale University, New Haven, CT, USA

Correspondence: thomas.pollard@yale.edu

structure (Robinson et al, 2001) toward the short-pitch actin helix in the branch junction (Rouiller et al, 2008). This shift was about 2-fold larger when either T237 or T238 of Arp2 was phosphorylated and/or R105 of ARPC4 was replaced with alanine, although the changes during the simulation time explored were far short of the 30 Å displacement of these subunits in the branch junction. As predicted by the MD simulation results, substituting alanine for R105 and R106 partially activated the purified Arp2/3 complex without an NPF (Narayanan et al, 2011).

These articles make a strong case that Arp2 phosphorylation relieves autoinhibitory interactions between two threonines or a tyrosine of Arp2 and three arginines on the Arp3 and ARPC4 subunits, inducing a conformational change that partially activates the complex. However, other evidence indicate that phosphorylation at the three proposed sites is not essential for some activities of the Arp2/3 complex. For example, replacing the two threonines and the tyrosine with alanine in *Dictyostelium* Arp2 slowed development but not pseudopod extension in a chemotaxis assay (Choi et al, 2013). Purified *B. taurus* Arp2/3 complex can nucleate one actin branch per complex under ideal conditions in vitro (Higgs et al, 1999), but electron density maps of a 2.0 Å crystal structure of bovine Arp2/3 complex from the same preparation (Robinson et al, 2001) had no density corresponding to phosphorylation of Y202, T237, or T238 (Fig S1A). Therefore, further study was needed to determine if phosphorylation at the proposed sites regulates the Arp2/3 complex.

Here, we explore the effect of Arp2 phosphorylation on Arp2/3 complex activity in the fission yeast *Schizosaccharomyces pombe*, where efficient homologous recombination facilitates making mutations in the genome (Grimm et al, 1988; Bahler et al, 1998), and quantitative fluorescence microscopy assays are available to characterize actin assembly during endocytosis (Berro & Pollard, 2014a). The proposed phosphorylation sites are conserved in *S. pombe* as Y198, T233, and T234 (Fig S1B). We made Arp2 mutations in the fission yeast genome that block phosphorylation or mimic constitutive phosphorylation at these three sites. Arp2 is essential for viability in *S. pombe* (Morrell et al, 1999) and, therefore, any mutations that inactivate the Arp2/3 complex should be lethal. We determined whether strains with each Arp2 mutation were viable and measured the rate of actin patch formation in each viable strain to investigate how phosphorylation at each site would affect Arp2/3 complex activity.

## Results and Discussion

### Detecting Arp2 phosphorylation with mass spectrometry

Mass spectrometry of the Arp2/3 complex purified in the presence of phosphatase inhibitors revealed phosphorylation of Arp2 at Y198 and Y218, previously unexplored but widely conserved sites in eukaryotes (Fig S1C and D). We did not detect phosphorylation of T233 or T234. This does not rule out their phosphorylation in vivo, since phosphorylation of Arp2 at Y198 and Y218 was lost in the *S. pombe* Arp2/3 complex purified without phosphatase inhibitors. It is possible that the addition of phosphatase inhibitors was insufficient to prevent dephosphorylation at T233 and T234.

### Generation and viability of Arp2 mutant fission yeast strains

To determine the effect of phosphorylation at the sites identified by LeClaire et al on Arp2/3 complex activity in the fission yeast, we generated haploid *S. pombe* strains with all possible combinations of mutations that preclude or mimic phosphorylation at Y198, T233, and T234 (Fig 1A and Table S1). Alanine was used to preclude threonine phosphorylation, and aspartic acid to mimic it. To prevent and mimic phosphorylation at Y198, we used phenylalanine and glutamate, respectively, because of their greater structural similarity to tyrosine.

All strains with mutations preventing phosphorylation were viable, including the triple Arp2 Y198F/T233A/T234A mutant (Fig 1A), although Arp2/3 complex activity is necessary for the viability of fission yeast (Morrell et al, 1999). Furthermore, cells depending on the triple mutant Arp2 successfully assembled actin patches marked by the actin crosslinking protein Fim1-GFP (Figs 1C and 2D). These patches appeared similar to actin patches observed in *S. pombe* with wild-type Arp2 (Fig 1B). Therefore, Arp2 phosphorylation at Y198, T233, and T234 is not essential for Arp2/3 complex activity in *S. pombe*.

By contrast, when a C-terminal GFP tag was added to the Arp2 subunit with the Y198F/T233A/T234A triple mutation, the strain was not viable. This parallels the failure of Y202A/T237A/T234A Arp2-GFP to rescue lamellipodial assembly in *Drosophila* cells depleted of Arp2 (LeClaire et al, 2008).

Most strains with mutations mimicking phosphorylation of Y202, T233, and T234 in Arp2 were also viable, although the two strains with the T233D mutation displayed a growth defect at high temperatures (Figs 1A and S2B and C). However, the two strains with the T233D/T234D double mutation to mimic phosphorylation at both threonines (AE10 and AE14) were not viable (Fig 1A). We assume that the T233D and T234D substitutions mimic phosphothreonine in Arp2 as aspartic and glutamic acids mimic phosphoserine, phosphothreonine, and phosphotyrosine in other proteins (Dephoure et al, 2013). For example, the biochemical properties of cofilin with the S3D mutation are similar to those of phosphorylated cofilin (Blanchoin et al, 2000). Phosphomimetic substitutions are imperfect and can fail, especially at tyrosine residues because of the structural dissimilarity between phosphotyrosine and glutamate (Anthis et al, 2009). However, failure is most likely if the phosphorylated residue must fit precisely into a binding site, for example, on an adapter protein (Dephoure et al, 2013). This is not likely to pose an issue at T233 and T234, as they are located on the interior of the Arp2/3 complex (Robinson et al, 2001), and phosphorylation at these sites is proposed to disrupt binding interactions of Arp2 with Arp3 and ARPC4 (LeClaire et al, 2008).

### Effect of Arp2 mutations at proposed phosphorylation sites on actin patch assembly

We used quantitative fluorescence microscopy of live cells (Berro & Pollard, 2014a) to compare the rates at which wild-type and mutant Arp2/3 complexes assembled branched actin filaments in endocytic patches labeled with Fim1-GFP. Because individual actin patches vary in their total fluorescence and timing of events (Figs 2A and S3A), we used continuous alignment to average the assembly and disassembly curves of 27–115 patches (Fig S3B). We calculated

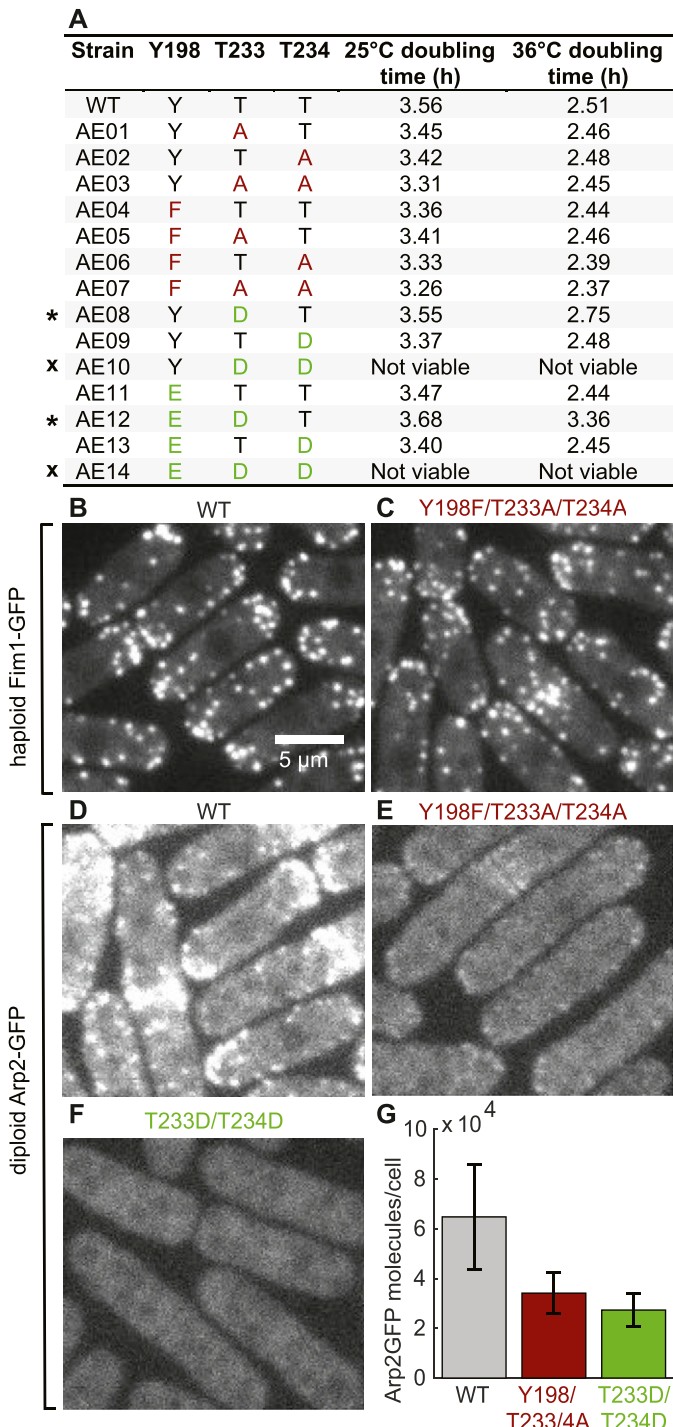

**A**

|   | Strain | Y198 | T233 | T234 | 25°C doubling time (h) | 36°C doubling time (h) |
|---|--------|------|------|------|------------------------|------------------------|
|   | WT     | Y    | T    | T    | 3.56 | 2.51 |
|   | AE01   | Y    | A    | T    | 3.45 | 2.46 |
|   | AE02   | Y    | T    | A    | 3.42 | 2.48 |
|   | AE03   | Y    | A    | A    | 3.31 | 2.45 |
|   | AE04   | F    | T    | T    | 3.36 | 2.44 |
|   | AE05   | F    | A    | T    | 3.41 | 2.46 |
|   | AE06   | F    | T    | A    | 3.33 | 2.39 |
|   | AE07   | F    | A    | A    | 3.26 | 2.37 |
| * | AE08   | Y    | D    | T    | 3.55 | 2.75 |
|   | AE09   | Y    | T    | D    | 3.37 | 2.48 |
| x | AE10   | Y    | D    | D    | Not viable | Not viable |
|   | AE11   | E    | T    | T    | 3.47 | 2.44 |
| * | AE12   | E    | D    | T    | 3.68 | 3.36 |
|   | AE13   | E    | T    | D    | 3.40 | 2.45 |
| x | AE14   | E    | D    | D    | Not viable | Not viable |

**B** WT

**C** Y198F/T233A/T234A

haploid Fim1-GFP

5 μm

**D** WT

**E** Y198F/T233A/T234A

diploid Arp2-GFP

**F** T233D/T234D

**G**

Arp2GFP molecules/cell

10 × 10⁴

WT   Y198/T233/4A   T233D/T234D

**Figure 1.  S. pombe strains with mutations of Arp2 residues Y198, T233, and T234.**
**(A)** Viability of haploid strains with mutations blocking or mimicking phosphorylation of residues Y198, T233, or T234 of Arp2, measured by growth of tetrads on YE5S plates at 25°C. Dark red: mutations blocking phosphorylation; light green: mutations mimicking it. * indicates a high-temperature growth defect; x indicates that the strain was not viable. **(B–F)** Sum projection of confocal fluorescence images (six z-sections with 0.6 μm spacing). **(B, C)** Haploid S. pombe expressing Fim1-GFP with (B) wild-type Arp2 or (C) Y198F/T233A/T234A mutant Arp2. Images taken from the first frame of actin patch time-lapse movies; both have identical contrast settings. **(D–F)** Diploid S. pombe strains with one copy of

the rate of actin patch assembly from the slope of the assembly phase of the averaged and aligned data (Fig S3C and D).

All of the viable haploid strains depending on the Arp2/3 complex with mutations that prevent or mimic phosphorylation of Arp2 residues Y198, T233, or T234 assembled and disassembled actin patches (Fig 2B–G), but all mutations of the three residues decreased the rate at which Fim1-GFP accumulated in the patches (Figs 2B–H and S3E). Cells depending on Arp2 with the triple Y198F/T233A/T234A mutation assembled actin patches slower than cells with T233A/T234A Arp2 (Fig 2C and D). Actin patches assembled slower in the strains with the Arp2-T233D phosphomimetic mutation, which displayed a growth defect at 36°C, than in those with the Arp2-T234D mutation, which grew near-normally (Fig 2E–G).

The subtle phenotypic changes that we observed in fission yeast strains with mutations of Arp2 phosphorylation sites are consistent with those detected in previous studies on other cells. On one hand, mutating Arp2 phosphorylation sites can produce strong phenotypes: Dictyostelium cells with Arp2 Y202F/T237A/T238A mutations develop very slowly when faced with starvation (Choi et al, 2013). On the other hand, these mutant cells had only subtle defects in speed and directionality during chemotaxis, a short-lived process much more akin to endocytosis than development. Y202A/T238A/T238A Arp2-GFP did not rescue lamellipodial assembly in cultured Drosophila cells depleted of Arp2 (LeClaire et al, 2008), but we found that the severity of this phenotype is likely because of the presence of the C-terminal GFP tag on the mutated Arp2.

If phosphorylation of Y198, T233, and T234 plays a role in regulating the Arp2/3 complex, then we would expect blocking and mimicking phosphorylation at these sites to have opposite effects on Arp2/3 complex activity. However, mutations that prevent phosphorylation and mutations that mimic it both modestly decreased the rate of endocytic actin patch assembly in yeast cells. Therefore, the Arp2 mutations we made compromised assembly by some other mechanism, as explored in the next section.

### Observations of mutated Arp2-GFP in diploid cells

To determine how lethal mutations such as T233D/T234D Arp2 and Y198F/T233A/T324A Arp2-GFP affect the behavior of the Arp2/3 complex in cells, we created diploid S. pombe strains with one copy of wild-type or mutant Arp2-GFP and one copy of untagged wild-type Arp2 to keep the cells alive. Individually, neither the GFP-tag nor the Y198F/T233A/T234A mutations rendered Arp2 nonfunctional: Haploid cells with untagged Y198F/T233A/T234A Arp2 built Fim1-GFP–labeled actin patches (Figs 1C and 2D), and wild-type Arp2-GFP was incorporated into patches in diploid cells (Fig 1D). However, most of the GFP-tagged Arp2 in diploids with either the Y198F/T233A/T234A mutations (Fig 1E) or the lethal T233D/T234D mutations (Fig 1F) localized to the cytoplasm. This explains why S. pombe strains with T233D/T234D mutations in Arp2 or Y198F/T233A/T234A mutations in Arp2-GFP are not viable.

The lethal Arp2 mutants may render the Arp2/3 complex nonfunctional by compromising its stability, rather than by modeling an

untagged wild-type Arp2 and one copy of (D) wild-type Arp2-GFP, (E) Y198F/T233A/T234A Arp2-GFP, or (F) lethal mutant T233D/T234D Arp2-GFP. All three images have identical contrast settings. **(G)** Measurements of Arp2-GFP molecules per cell in diploid strains (mean ± SD, n = 47–82).
Source data are available for this figure.

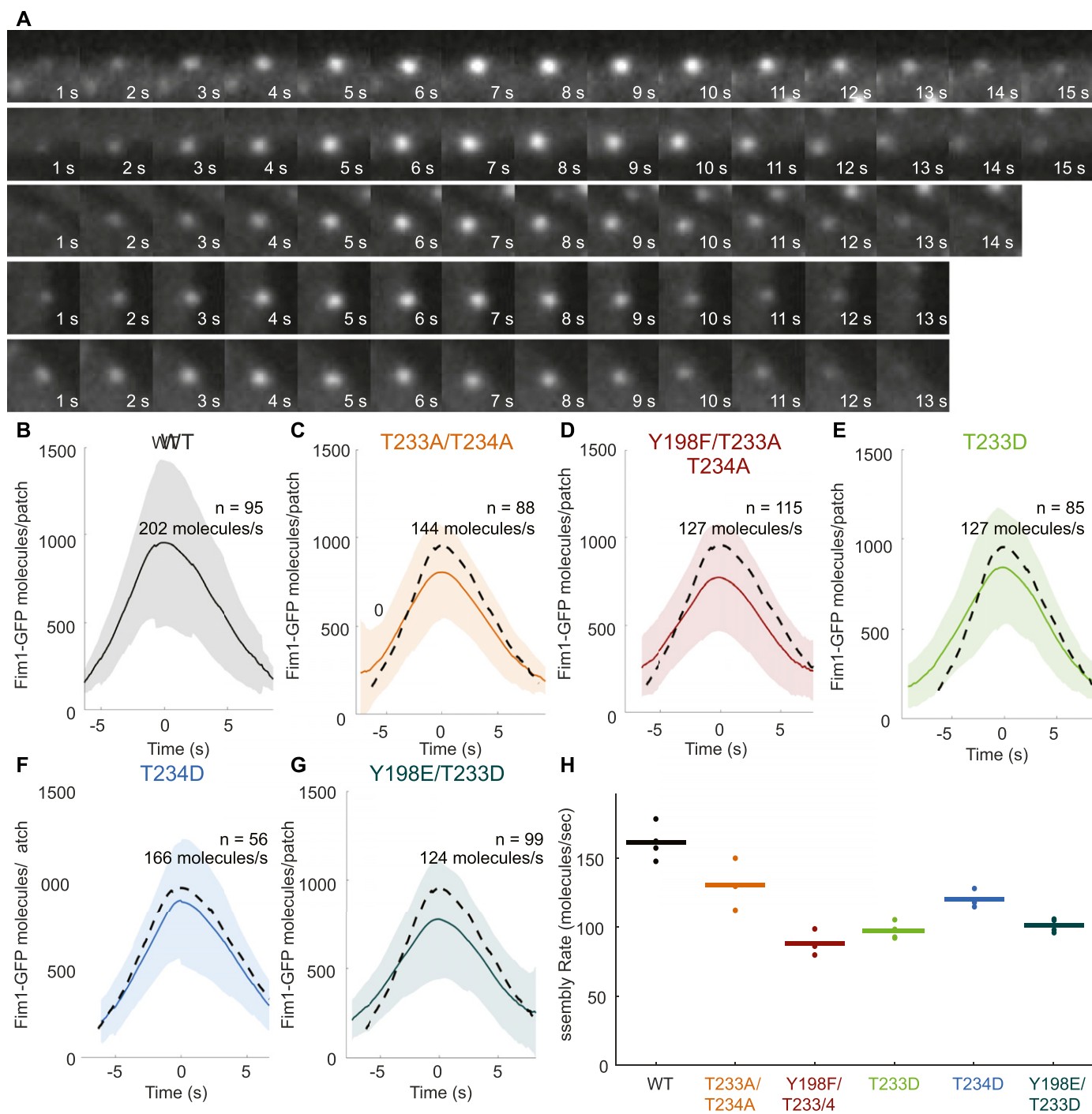

**Figure 2. Time course of actin patch assembly and disassembly by strains with Arp2 mutations either blocking or mimicking phosphorylation at Y198, T233, and T234.**
**(A)** Time series of fluorescence micrographs at 1 s intervals of individual actin patches in cells expressing Fim1-GFP, reconstructed from sum projections of six Z-sections. **(B–G)** Mean numbers of Fim1-GFP molecules over time in 56–115 aligned actin patch tracks from haploid *S. pombe* strains with (B) wild-type Arp2, (C–D) Arp2 with mutations blocking phosphorylation, and (E–G) Arp2 with phosphomimetic mutations. Shaded regions indicate standard deviations. Dashed lines represent mean numbers of molecules per patch from the wild-type strain in panel A. **(H)** Rates of patch assembly for 3–4 replicates (27–115 patches per replicate) of time-lapse movies of *S. pombe* with wild-type Arp2 and Arp2 with mutations blocking or mimicking phosphorylation. Points indicate assembly rates of each replicate; horizontal bars denote the mean assembly rates.

altered phosphorylation state. Wild-type Arp2 dissociates from a fraction of fission yeast Arp2/3 complex during purification (Nolen & Pollard, 2008), and mutating the proposed phosphorylation sites at the interface between Arp2 and Arp3/ARPC4 could further weaken the binding of Arp2 to its neighbors. Unassembled components of the Arp2/3 complex (Morrell et al, 1999) and other protein complexes, such as α- and β-spectrin (Woods & Lazarides, 1985) and WAVE regulatory complex (Kunda et al, 2003), are

degraded. Consistent with degradation of unassembled Arp2, quantitative microscopy of diploid strains revealed lower cellular levels of mutant Arp2-GFP than wild-type Arp2-GFP (Fig 1G). Furthermore, the Y198F/T233A/T234A Arp2/3 complex did not bind to a column with GST-tagged VCA (verprolin homology, central, acidic) domain of WASp and elute intact from the subsequent ion exchange column during purification attempts (Fig S2D). The reduced growth of T233 mutants at 36°C is also consistent with compromised protein–protein interactions causing the observed defects, as increased temperature could weaken these interactions further. Therefore, our data suggest that mutations of Y198, T233, and T234 compromise some aspect of Arp2/3 complex function, reflected in the rate of actin patch assembly (Fig 2).

### Effect of Y218 Arp2 mutations on actin patch assembly

Prior evidence indicates that the three proposed phosphorylation sites may not be the only residues at which phosphorylation regulates the Arp2/3 complex (LeClaire et al, 2008). The *Drosophila* kinase NIK phosphorylated some sites on Arp2 and ARPC2 (detected by autoradiography with $^{32}$P) and restored the ability of phosphatase-treated *Acanthamoeba* Arp2/3 complex to nucleate actin filaments in vitro (LeClaire et al, 2015). However, NIK also partially restored the nucleation activity of the Arp2/3 complex with Y202A/T237A/T238A Arp2, although it could not phosphorylate these three residues of Arp2

(LeClaire et al, 2015). One way to reconcile these observations is that NIK activates the Arp2/3 complex by phosphorylating an uninvestigated site on one of its subunits, possibly Arp2 or ARPC2.

Mass spectrometry revealed a previously unidentified phosphorylation site on the fission yeast Arp2 subunit: the conserved residue Y218 (Figs 3A and S1D). Haploid fission yeast strains with Y218 replaced by either phenylalanine or glutamate were viable and built branched actin patches, which appeared similar to patches constructed by the wild-type Arp2/3 complex (Fig 3B–D). However, both blocking and mimicking phosphorylation of Y218 subtly decreased Arp2/3 complex activity (Figs 3E–H and S3F). Therefore, phosphorylation at Y218 is not essential for the Arp2/3 complex to be active, and if the Y218E substitution effectively mimicked phosphotyrosine, then any effects due to mutations at this site resulted from compromising the protein structure. Future research should consider if phosphorylation at undetected sites on Arp2 and/or on another subunits activates the Arp2/3 complex.

## Materials and Methods

### Purification of the Arp2/3 complex

Fission yeast was grown in 4–8 liters of YE5S culture medium until the $OD_{595}$ of a 1:5 dilution was 0.3–0.5 (1.5–2.5 × $10^7$ cells/ml), at which

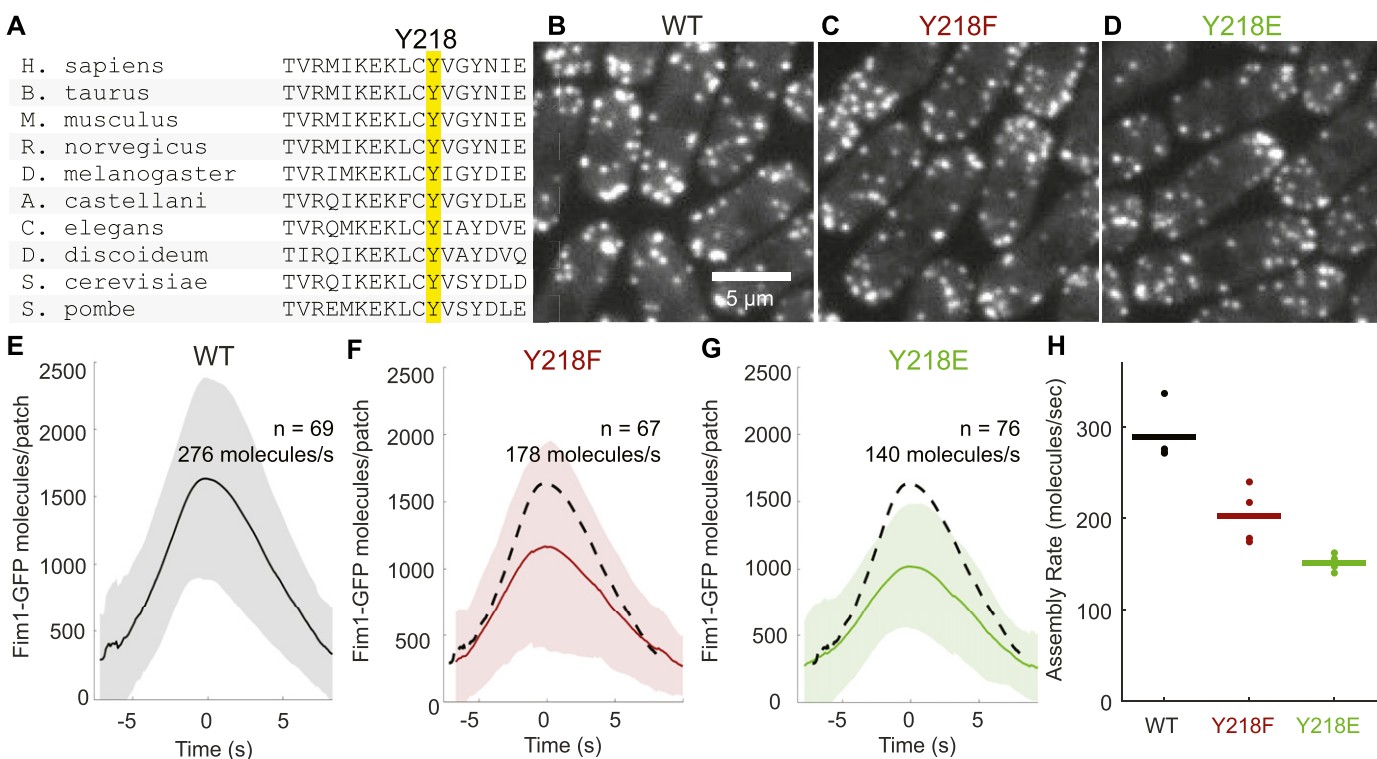

**Figure 3. Phosphorylation at conserved Arp2 residue Y218 is not essential for Arp2/3 complex activity.**
**(A)** Sequence alignment of *S. pombe* Arp2 residues 208–224 with homologous regions of Arp2 in nine other eukaryotes. Y218 and homologous residues are highlighted. **(B–D)** Sum projection of confocal fluorescence images (six Z-sections with 0.6 µm spacing) of haploid *S. pombe* endogenously expressing Fim1-GFP with (B) wild-type Arp2, (C) Arp2 Y218F, or (D) Arp2 Y218E. Images taken from the first frame of actin patch time-lapse movies; all have identical contrast settings. **(E–G)** Mean numbers of Fim1-GFP molecules over time in 67–76 aligned actin patch tracks from (E) wild-type cells, (F) Y218F Arp2, or (G) Y218E Arp2. Dashed lines represent mean number of molecules per patch from wild-type Arp2 alignment in panel E. **(H)** Rates of patch assembly for 3–4 replicates (24–76 patches per replicate) of wild-type and Y218 Arp2 mutants. Points indicate assembly rates of each replicate; horizontal bars denote the mean assembly rate.

point a further 70 g/l YE5S powder was added and cells were allowed to resume growing. Cultures were pelleted and frozen when the OD$_{595}$ of a 1:10 dilution was 0.55–0.65 (5.5–6.5 × 10$^7$ cells/ml). The cells were thawed, lysed using a microfluidizer, and centrifuged, and the Arp2/3 complex was purified from the soluble fraction by chromatography on a GST-VCA affinity column (GE Healthcare Life Sciences), a MonoQ 5/50 ion exchange column (GE Healthcare Life Sciences), and a HiLoad Superdex 200 16/60 gel purification column (GE Healthcare Life Sciences) (Ti et al, 2011). Roche cOmplete EDTA-free protease inhibitor tablets (MilliporeSigma) were added as the pellet thawed and periodically during initial centrifugation and dialysis steps. Roche PhosSTOP phosphatase inhibitor tablets (MilliporeSigma) were added alongside protease inhibitors in some preparations. The concentration of the purified Arp2/3 complex was measured by absorption at 290 nm ($\varepsilon$ = 139,030 M$^{-1}$·cm$^{-1}$).

### Mass spectrometry of the *S. pombe* Arp2 subunit

Purity was confirmed by SDS–PAGE and staining the seven bands corresponding to the Arp2/3 complex subunits with Coomassie blue. The Arp2 (44 kD) band was cut from the gel and digested with 6.67 $\mu$g/ml trypsin for 18 h at 37°C in 10 mM NH$_4$HCO$_3$ buffer. Digested samples were analyzed using liquid chromatography–tandem mass spectrometry on a Waters/Micromass AB QSTAR Elite mass spectrometer (Stone & Williams, 2009) by the Yale Mass Spectrometry and Proteomics Resource of the W.M. Keck Foundation. Phosphopeptides were identified using the Mascot algorithm (Hirosawa et al, 1993) through the Yale Protein Expression Database (Colangelo et al, 2015). Mass spectra from fragmentation of each identified phosphopeptide were checked (Fig S1C and D) to ensure that fragments identified were sufficient to distinguish phosphorylation from other chemical modifications.

### Generation of haploid and diploid *S. pombe* Arp2 mutant strains

We first used a two-step homologous recombination process (Grimm et al, 1988; Bahler et al, 1998) to create point mutations in *arp2* of diploid *S. pombe* strains because *arp2* is an essential gene. Haploid *S. pombe* strains with complementary adenine auxotrophy mutations *ade6-M210* and *ade6-M216* (Liang et al, 1999; Hoffman et al, 2016) were crossed to create diploid *S. pombe*, which we selected using adenine-deficient media. One copy of the wild-type *arp2* gene in diploid *S. pombe* lacking *ura4* was replaced by homologous recombination with an *ura4*$^+$ selectable marker, and transformed cells were selected on EMM5S-ura plates. The *ura4*$^+$ marker was replaced with mutant *arp2* or *arp2-GFP* adjacent to the *kan*$^R$ resistance marker, and transformants were selected on YE5S-G418 plates.

To create haploid yeast strains, we induced diploid yeast to sporulate on SPA5S medium for 24–48 h and then dissected tetrads on YE5S plates. Tetrads were grown for several days at 25°C and then replica-plated onto YE5S-G418 plates to identify progeny with the *arp2* mutation and the *kan*$^R$ allele. All mutations were confirmed by DNA sequencing. A mutation was judged to be lethal if only two of four spores grew from each tetrad, corresponding to the two haploid progeny with the wild-type *arp2* allele. These progeny did not possess the *kan*$^R$ resistance allele and did not grow on YE5S-G418.

Haploid yeast strains were crossed with *fiim1-GFP* haploid yeast and tetrads dissected to create *fim1-GFP*–labeled haploid strains.

### Imaging *S. pombe* cells

*S. pombe* strains were grown in YE5S media in the log phase (OD$_{595}$ < 0.8) for 36 h. The cells were harvested at OD$_{595}$ 0.4–0.7 (4–7 × 10$^6$ cells/ml), washed twice with EMM5S media, and mounted on EMM5S 2% agarose pads. Images were acquired at room temperature with an Olympus IX-71 microscope with a 100×/NA 1.4 Plan Apo lens (Olympus) and an Andor CSU-X1 spinning disk confocal system with an iXON-EMCCD camera (Andor Technology). Fluorescence was excited using a Coherent OBIS 488-nm LS 20-mW laser with adjustable power and an internal power meter. Time-lapse movies of endocytic actin patches were acquired with Andor iQ2 software (Andor Technology); other images were acquired with the $\mu$Manager 1.4 plugin (Stuurman et al, 2010) for ImageJ (Schneider et al, 2012).

### Generating the calibration curve and calculating number of molecules per cell

21 Z-sections with 0.6 $\mu$m spacing were taken of eight yeast strains: one nonfluorescent strain (FY528) and seven endogenously expressing one of the following GFP-tagged proteins: Myo2p, Ain1p, Acp2p, ARPC5, Arp2p, Arp3p, and Fim1p (Table S1; Wu & Pollard, 2005).

To generate a calibration curve automatically (Wu & Pollard, 2005), we designed the ImageJ macro AutoCalibrationCurve (https://github.com/tdpollardlab/AutoCalibrationCurve). Sum projections of fluorescence image Z-stacks were corrected for camera noise and uneven illumination and segmented to isolate individual *S. pombe* cells. The total fluorescence intensity per cell in each strain was measured and plotted against the number of fluorescent molecules per cell, which was previously obtained by quantitative immunoblotting (Wu & Pollard, 2005). The slope of the best-fit line yielded the relationship between the total fluorescence intensity in a given intracellular region and the number of fluorescent molecules it contained (Fig S4B).

For image segmentation, the AutoCalibrationCurve macro used the Mitotic Analysis and Recording System (MAARS) (Li et al, 2017) plugin for ImageJ. The MAARS plugin automatically circled cells in sum projections of fluorescence images using 21 bright-field Z-sections spaced 0.6 $\mu$m apart (Fig S4A). We corrected for autofluorescence by subtracting the total fluorescence intensity per cell in the nonfluorescent strain.

We used the resulting calibration curve (Fig S4B) and the AutoCalibrationCurve software to calculate the total number of fluorescent molecules per cell in each diploid strain. Autofluorescence was corrected by subtracting the total fluorescence intensity per cell in the diploid strain without any GFP (AE15D).

### Acquiring actin patch tracks

We took 3–4 time-lapse movies with 1-s intervals for 60 s of each haploid yeast strain expressing endogenous Fim1-GFP. Each image consisted of six Z-sections separated by 0.6 $\mu$m collected with 5 mW laser power (as measured at the emission source; approximate

power at the sample was 0.13 mW). An original MATLAB (MathWorks) script corrected the images for uneven illumination, camera noise, and photobleaching (Fig S4C–H). Endocytic actin patches in corrected time-lapse movies were identified and tracked using the Fiji (Schindelin et al, 2012) plugin PatchTrackingTools (Berro & Pollard, 2014b). Identified patches were screened manually to reject records missing the beginning of assembly or the end of disassembly, or where two patches overlapped.

All accepted patch tracks within each time-lapse movie were aligned and their total fluorescence intensities (integrated density) were averaged using temporal super-resolution realignment (Fig S3B; Berro & Pollard, 2014a). The calibration curve was used to derive the number of molecules in each actin patch from its total fluorescence intensity. The resulting number of Fim1-GFP molecules in the average actin patch was plotted as a function of time, generating one assembly/disassembly curve per time-lapse movie. The movie with the largest sample of patches was chosen for example figures.

To measure the rate at which actin patches assembled, we created an original MATLAB script (https://github.com/tdpollardlab/findLinearRegions) to fit straight lines to the most linear 3.7 s regions of the assembly and disassembly phases (Fig S3B and C). The mean assembly rate for each strain was determined by averaging the assembly rates observed for 3–4 averaged assembly/disassembly curves (Fig S3C and D). Patch assembly/disassembly curves were plotted in MATLAB using color-blind–safe colors (Martin Krzwinski, http://mkweb.bcgsc.ca/biovis2012/color-blindness-palette.png). Shaded regions indicating SD were generated using the shadedErrorBar MATLAB script (Rob Campbell, https://www.mathworks.com/matlabcentral/fileexchange/26311-raacampbell-shadederrorbar).

### Automatic photobleaching correction of yeast time-lapse movies

We automated the photobleaching correction process with the original MATLAB script PhotoBleachingCorrector (https://github.com/tdpollardlab/PhotoBleachingCorrector). Time-lapse movies were first corrected for camera noise and uneven illumination. Poorly illuminated areas were then identified and cropped using the best-fitting two-dimensional Gaussian function to the uneven illumination correction image. To identify intracellular regions, each of the first 10 frames of each time-lapse movie was thresholded by brightness. Pixels brighter than the threshold in eight of the 10 frames were assumed to be intracellular (Fig S4E).

To determine the threshold used, all pixels in the cropped image were binned according to brightness. The resulting histogram consisted of two populations: one of intracellular and one of background pixels. Both populations were expected to have a normal brightness distribution, and the histogram was, therefore, fit to a two-term Gaussian function. Background pixels were identified by fitting a one-term Gaussian function to the region of the histogram preceding the peak. The threshold was set at the intensity bin for which 95% of pixels were predicted to be intracellular (Fig S4D).

The median fluorescence of the intracellular pixels was calculated for each frame of the time-lapse movie, plotted, and fit to a double-exponential decay function; it is likely that the first exponential term corresponds to photobleaching of autofluorescence

and the second to bleaching of GFP (Fig S4F). The decay function was normalized to start at one, and time-lapse movies were corrected by dividing each frame by the corresponding value of the normalized decay function (Fig S4G and H).

### *S. pombe* growth in liquid culture

Cells were grown in YE5S with shaking for 36 h in the log phase at 25°C. Cultures were diluted to $OD_{595}$ 0.1 and incubated with shaking at 25°C or 36°C. The $OD_{595}$ was measured every 1.5–2.5 h for 14 h, and then again at approximately 22 and 26 h (Fig S2A and B). The $\log_2$ of the measured OD values was plotted as a function of time, and the slope of the best-fit line to the first eight points (i.e., the log phase) was used to calculate the growth rate for each strain (Fig 1A).

### *S. pombe* serial dilution growth assays

Cells were grown in YE5S for 36 h in the log phase with shaking. Cultures were harvested at an $OD_{595}$ of 0.4–0.6 (35–65 × $10^5$ cells/ml) and diluted to $OD_{595}$ 0.1. After 1:10 serial dilutions, 10 $\mu$l samples were plated on YE5S and incubated for 24–48 h at 32°C or 36°C (Fig S2C).

## Data Availability

Source data for actin patch time-lapse movies (raw data and photobleaching-corrected sum projections), along with image stacks used to generate the calibration curve, are available in the BioStudies database (http://www.ebi.ac.uk/biostudies) under the accession number S-BSST205.

## Online Supplemental Material

Fig S1A and B shows the location of the previously proposed phosphorylation sites Y198, T233, and T234 on Arp2 and that they are widely conserved in eukaryotes. Fig S1C and D depicts mass spectra from fragmentation of peptides containing phosphorylation at Y198 and Y218. Fig S2 demonstrates the effect of mutations at Y198, T233, and T234 on (A–C) fission yeast growth and (D) on yield when the Arp2/3 complex is purified using a GST-VCA column. Fig S3A–D shows tracking and temporal super-resolution realignment of Fim1-GFP–labeled actin patches with PatchTrackingTools and (E, F) contains the assembly rates obtained from the aligned data. Fig S4A and B shows how images are segmented and how the total intracellular fluorescence of each strain is used to generate the calibration curve. Fig S4C–H demonstrates photobleaching correction of a sample time-lapse movie containing multiple Fim1-GFP–labeled actin patches.

## Supplementary Information

## Acknowledgments

The authors thank Dr. Julien Berro for his help with the automated patch tracking and analysis software, Dr. Tong Li for support with the MAARS segmentation software, Helen Sun for reading the manuscript, Jean Kanyo for technical assistance with mass spectrometry, Dr. Moon Chatterjee and Nandan Pandit for advice regarding Arp2/3 complex purification, and Dr. Samantha E.R. Dundon for invaluable assistance with yeast culture techniques, microscopy, and editing. Research reported in this publication was supported by National Institute of General Medical Sciences of the National Institutes of Health under award number R01GM026338. The content is solely the responsibility of the authors and does not necessarily represent the official views of the National Institutes of Health. This research was also supported by fellowships to Alexander Epstein from the Arnold and Mabel Beckman Foundation and from the Yale University Office of Science and Quantitative Reasoning.

### Author Contributions

AE Epstein: conceptualization, data curation, software, formal analysis, validation, investigation, visualization, methodology, and writing—original draft, review, and editing
S Espinoza-Sanchez: conceptualization, data curation, formal analysis, investigation, and methodology.
TD Pollard: conceptualization, resources, supervision, funding acquisition, methodology, project administration, and writing—original draft, review, and editing.

### Conflict of Interest Statement

The authors declare that they have no conflict of interest.

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
