## [Reviewer comments · Life Science Alliance]

Life Science Alliance

Phosphorylation of Arp2 is not essential for Arp2/3 complex activity in fission yeast

Alexander Epstein, Sofia Espinoza-Sanchez, and Thomas Pollard
DOI: 10.26508/lsa.201800202

Corresponding author(s): Thomas Pollard, Yale University

Review Timeline:	Submission Date:	2018-09-27
	Editorial Decision:	2018-09-28
	Revision Received:	2018-10-10
	Editorial Decision:	2018-10-10
	Revision Received:	2018-10-12
	Accepted:	2018-10-12

Scientific Editor: Andrea Leibfried

Transaction Report:

Please note that the manuscript was previously reviewed at another journal and the reports were taken into account in the decision-making process at Life Science Alliance. Since the original reviews are not subject to Life Science Alliance's transparent review process policy, the reports and author response cannot be published.

September 28, 2018

Re: Life Science Alliance manuscript #LSA-2018-00202-T

Dr. Thomas D Pollard
Yale University
Molecular, Cellular and Developmental Biology
Kline Biology Tower 222
266 Whitney Avenue
New Haven, CT 06511-8112

Dear Dr. Pollard,

Thank you for transferring your manuscript entitled "Phosphorylation of Arp2 is not essential for Arp2/3 complex activity in fission yeast" to Life Science Alliance. The manuscript was assessed by expert reviewers at another journal before, and the editors transferred those reports to us with your permission.

The reviewers think that your data are of high quality and of value to the field. They noted, however, that the controversy regarding the role of Arp2/3 complex phosphorylation for complex function is not entirely settled, because such role could be context- or organism-dependent. Based on the reviewer reports already at hand, we would be happy to publish your work in Life Science Alliance pending minor revision. We would expect a point-by-point response to the specific criticisms raised by the reviewers and accordingly changes to the data representation and inclusion of a statistical test for Fig2. We would further appreciate if you could provide expression level analysis for Arp2/3 complex components as requested by reviewers #1 and #3.

Thank you for this interesting contribution to Life Science Alliance. We are looking forward to receiving your revised manuscript.

Sincerely,

Andrea Leibfried, PhD
Executive Editor
Life Science Alliance
Meyerhofstr. 1
69117 Heidelberg, Germany
t +49 6221 8891 502

- A letter addressing the reviewers' comments point by point.
- An editable version of the final text (.DOC or .DOCX) is needed for copyediting (no PDFs).
- High-resolution figure, supplementary figure and video files uploaded as individual files: See our detailed guidelines for preparing your production-ready images, <http://life-science-alliance.org/authorguide>
- Summary blurb (enter in submission system): A short text summarizing in a single sentence the study (max. 200 characters including spaces). This text is used in conjunction with the titles of papers, hence should be informative and complementary to the title and running title. It should describe the context and significance of the findings for a general readership; it should be written in the present tense and refer to the work in the third person. Author names should not be mentioned.

B. MANUSCRIPT ORGANIZATION AND FORMATTING:

Full guidelines are available on our Instructions for Authors page, <http://life-science-alliance.org/authorguide>

October 10, 2018

RE: Life Science Alliance Manuscript #LSA-2018-00202-TR

Dr. Thomas D Pollard
Yale University
Molecular, Cellular and Developmental Biology
Kline Biology Tower 222
266 Whitney Avenue
New Haven, CT 06511-8112

Dear Dr. Pollard,

Thank you for submitting your revised manuscript entitled "Phosphorylation of Arp2 is not essential for Arp2/3 complex activity in fission yeast", and for sending a point-by-point response to the criticisms raised during peer review elsewhere.

I appreciate the introduced changes and the reasoning for not analyzing the amount of Arp2/3 complex expressed in the cells by other means. Epitope tags affect the viability of some mutants and there are no specific antibodies available that do not cross-react with other *S. pombe* proteins, thus not allowing to test expression levels by western blot. I also value your response to the request to include a statistical analysis for the differences in the number of molecules per patch for different Arp2 mutants in Fig 2G, outlining that the number of separate experiments is not large enough to do so.

In light of your response and the changes introduced in the manuscript, we would be happy to publish your paper in Life Science Alliance pending final revisions necessary to meet our formatting guidelines:

- please add callouts to the following figures in the manuscript text:

Fig 1B,D

Fig 2B,C,E,F,G

Fig 3,B,C,D,E,F,G

Fig S4B

A. FINAL FILES:

-- High-resolution figure, supplementary figure and video files uploaded as individual files: See our detailed guidelines for preparing your production-ready images, <http://life-science-alliance.org/authorguide>

B. MANUSCRIPT ORGANIZATION AND FORMATTING:

Full guidelines are available on our Instructions for Authors page, <http://life-science-alliance.org/authorguide>

Sincerely,

Andrea Leibfried, PhD
Executive Editor
Life Science Alliance

Meyerhofstr. 1
69117 Heidelberg, Germany
t +49 6221 8891 502
e a.leibfried@life-science-alliance.org
www.life-science-alliance.org

RE: Life Science Alliance Manuscript #LSA-2018-00202-TRR

Dr. Thomas D Pollard
Yale University
Molecular, Cellular and Developmental Biology
Kline Biology Tower 222
266 Whitney Avenue
New Haven, CT 06511-8112

Dear Dr. Pollard,

Thank you for submitting your Research Article entitled "Phosphorylation of Arp2 is not essential for Arp2/3 complex activity in fission yeast". It is a pleasure to let you know that your manuscript is now accepted for publication in Life Science Alliance. Congratulations on this interesting work.

The final published version of your manuscript will be deposited by us to PubMed Central (PMC) as soon as we are allowed to do so, the application for PMC indexing has been filed. You may be eligible to also deposit your Life Science Alliance article in PMC or PMC Europe yourself, which will then allow others to find out about your work by Pubmed searches right away. Such author-initiated deposition is possible/mandated for work funded by eg NIH, HHMI, ERC, MRC, Cancer Research UK, Telethon, EMBL.

Please also see:

<https://www.ncbi.nlm.nih.gov/pmc/about/authorms/>

<https://europepmc.org/Help#howsubmanu>

DISTRIBUTION OF MATERIALS:

Again, congratulations on a very nice paper. I hope you found the review process to be constructive and are pleased with how the manuscript was handled editorially. We look forward to future exciting submissions from your lab.

Sincerely,

Andrea Leibfried, PhD

Executive Editor
Life Science Alliance
Meyerhofstr. 1
69117 Heidelberg, Germany
t +49 6221 8891 502
e a.leibfried@life-science-alliance.org
www.life-science-alliance.org